# Finely Tunable Carbon Nanofiber Catalysts for the Efficient Production of HMF in Biphasic MIBK/H_2_O Systems

**DOI:** 10.3390/nano14151293

**Published:** 2024-07-31

**Authors:** Charf Eddine Bounoukta, Cristina Megías-Sayago, Nuria Rendón, Fatima Ammari, Miguel Angel Centeno, Svetlana Ivanova

**Affiliations:** 1Departamento de Química Inorgánica e Instituto de Ciencia de Materiales de Sevilla, Centro Mixto CSIC-Universidad de Sevilla, 41092 Sevilla, Spain; charf-eddine.bounoukta@univ-littoral.fr (C.E.B.); centeno@icmse.csic.es (M.A.C.); 2Laboratoire de Génie des Procédés Chimiques-LGPC, Département de Génie des Procéés, Faculté de Technologie, Université Ferhat Abbas Setif-1, Setif 19000, Algeria; ammarifatima@yahoo.fr; 3Departamento de Química Inorgánica e Instituto de Investigaciones Químicas, Centro Mixto CSIC-Universidad de Sevilla, 41092 Sevilla, Spain; cmegias@us.es (C.M.-S.); nrendon@us.es (N.R.)

**Keywords:** HMF production, fructose dehydration, CNF functionalization, tandem Brønsted–Lewis sites

## Abstract

This work proposes catalytic systems for fructose dehydration to 5-hydroxymethylfurfural using a series of functionalized carbon nanofibers. The catalysts were synthesized via finely selected covalent grafting in order to include a variety of functionalities like pure Bronsted acid, tandem Brønsted/Lewis acid, and tandem Lewis acid/Lewis base catalysts. After the characterization and evaluation of acidity strength and the amount of acid centers, the catalyst series was screened and related to the product distribution. The best-performing catalyst was also used to optimize the reaction parameters in order to achieve 5-hydroxymethylfurfural yields rounding at 60% without significant humin formation.

## 1. Introduction

Sugars derived from lignocellulose, which have an annual production of 1 × 10^11^ tons/year, are currently the most important renewable source used to produce fuels and chemicals that are able to replace fossil fuels and industrial commodities [1,2,3,4]. The hydrolysis and dehydration of sugars produces, among others, 5-hydroximethylfurfural (HMF), a bifunctional molecule possessing a furan ring and an aldehyde group, both allowing the production of a diverse array of chemicals of highly added value [5,6]. Although desirable, direct cellulose conversion to HMF suffers from reactive insolubility in water or other solvents, pushing the technology to a two-step process: hydrolysis to glucose followed by further transformation to HMF [7]. Glucose is abundant in nature and can be transformed to HMF via a few steps requiring a high energy barrier and distinct active sites [8]. One of the paths proceeds through glucose isomerization to fructose and subsequent rapid dehydration to HMF [9,10]. That is why fructose is frequently chosen as the initial substrate to screen catalysts for dehydration under acidic conditions at low temperatures. Many considerable efforts have been focused on the production of HMF from fructose with high yields [11,12,13]. However, the commercial viability of this approach faces one main challenge: the minimization of undesired products (humins, levulinic, and formic acid) and the high cost of their separation [14]. The higher energy barrier and side reactions in aqueous media necessitate the use of organic solvents, increasing the price of the processes, which now require a separation procedure, which is a high-energy and time-consuming operation [15]. 

High HMF production from fructose is usually obtained using Brønsted acidic sites in the presence of dimethyl sulfoxide (DMSO) or ionic liquids, with the latter being used as co-catalysts and/or solvents [16]. DMSO does not seem the perfect choice of solvent due to its high boiling point and difficult distillation/rectification [17,18]. In this sense, the use of ionic liquids could solve the problem but, unfortunately, organic salts are expensive and complex to recycle [19]. The use of biphasic reaction systems that include two solvents, such as THF/H_2_O, MIBK/H_2_O, and butanol/H_2_O, etc., is an alternative and could improve the environmental and economic profile since they permit generous HMF production and immediate isolation in a single step [20,21,22,23].

HMF production can also be improved by applying a new class of hybrid materials with a variety of acidic sites incorporated via covalent functionalization on the surface of different solids [24,25,26,27,28]. These types of catalysts meet the principles of green chemistry and allow for control of the fructose reaction to HMF [29]. In our previous work, we have successfully used *p*-toluene sulfonic acid-functionalized carbon materials as catalysts, yielding significant HMF production [30,31]. Nevertheless, the overall process suffers from leaching of the non-covalently attached active sites, causing a short life catalyst cycle; therefore, it has limited application in potential biorefinery reactions [32]. That is why in this study we opted for a covalent chemical functionalization of stable nanostructures, such as carbon nanofibers (CNFs) [33]. This functionalization should create a uniform organic–inorganic structure that inhibits side reactions and minimizes active site leaching [34,35,36,37]. As a functionalizing component, we have chosen ionic liquid phase and organosilane moieties immobilized on a solid surface. The use of such materials should allow for an increase in dehydration reaction activity via the generation of particular hydrophilic properties and tandem sites on the carbon surface. This functionalization was chosen carefully to permit different functional groups/active species, such as sulfurand nitrogen containing groups of the corresponding organosilane and imidazolium ILs moieties. A deeper understanding of the role of the grafted molecule should also permit more successful fine-tuning of the strength and character of the active sites to effectively limit the side reactions and increase HMF production. 

Hence, the main purpose of this work is to graft some ILs and organosilane moieties over carbon nanofibers in order to produce hybrid catalysts for fructose dehydration in biphasic MIBK/H_2_O media. In addition, a systematic study of the reaction conditions has been carried out for the estimation of the optimal reaction conditions such as temperature, time, MIBK/H_2_O ratio, initial fructose concentration, and catalyst stability. A tentative mechanism of fructose dehydration over the hybrids is also proposed. 

## 2. Experimental

For the preparation of the samples, commercially available carbon nanofibers (CNFs) (GANF13, Grupo Antolin Ingeniería, Burgos, Spain) were used after pretreatment with concentrated nitric HNO_3_ (37%) or sulfuric acid H_2_SO_4_ (96%) at 60 °C or 150 °C, respectively, for 18 h in a round-bottomed flask equipped with reflux and magnetic agitation. The resulting solids were washed abundantly with water until reaching a neutral pH and dried overnight at 60 °C. The final CNF-O (nitric acid) and CNF-S (sulfuric acid) catalysts were ground in a mortar and used directly. 

For the grafting of organosilanes with amino or mercapto functionalities, 2 g of the above-described CNF-O sample was suspended in 50 mL of toluene and reacted with 1.5 mL of amino- or mercapto-propyltrimethoxysilane (APTMS or MPTMS) at 70 °C under a nitrogen atmosphere using standard Schlenk-type techniques, overnight. The obtained solids were filtered, washed three times with toluene, and dried at 100 °C overnight. The samples received the labels CNF-APTMS and CNF-MPTMS for amino- and mercaptosilane functionalization, respectively.

### 2.1. Preparation of CNF-SO_3_HPTMS

Further, the CNF-MPTMS hybrid was contacted with 40 mL of hydrogen peroxide solution (3/1 *v*/*v* H_2_O_2_/H_2_O) for 12 h at room temperature to transform the mercapto functionality in a sulfonic one. The obtained sample (CNF-SO_3_HPTMS) was filtered, washed with ethanol, and dried overnight at 60 °C.

For the immobilization of the ionic liquid onto the CNF surface, the oxidized CNF-O sample (2 g) was reacted with thionyl chloride (SOCl_2_) under nitrogen at 70 °C during 24 h. The excess of SOCl_2_ was removed by washing with anhydrous tetrahydrofuran (THF) and after drying, the solid was reacted with 3-chloro-1-propanol at 120 °C for 24 h under reflux in nitrogen. The excess of 3-chloro-1-propanol was removed by washing with dichloromethane and reacted after drying with 1-methyl imidazole under reflux at 95 °C during 24 h to form the final immobilized ionic liquid CNF catalyst, labeled as CNF-ILs. 

### 2.2. Catalytic Tests 

In total, 40 mg of catalyst was placed into a 50 mL Schlenk reactor equipped with a Young valve and magnetic stirrer and dispersed in a solution of 180 mg of fructose in 12 mL MIBK/H_2_O (1/5 *v*/*v*) mixture. After a nitrogen purge, the mixture was reacted at the desired temperature and time under continuous stirring of 600 rpm. After reaction, the liquid suspension was quenched in an ice bath, microfiltered with a 0.45 μm Nylon membrane, and analyzed by HPLC using 0.005 M H_2_SO_4_ as the mobile phase and a HiPlex-H column at 40 °C.

The catalytic activity results were expressed in terms of fructose conversion, HMF yield, and product selectivity, defined as follows:(1)$$\mathrm{Fructose}\;\mathrm{conversion}\;\left(\%\right)=\frac{\mathrm{moles}\;\mathrm{of}\;\mathrm{reacted}\;\mathrm{fructose}}{\mathrm{moles}\;\mathrm{of}\;\mathrm{initial}\;\mathrm{fructose}}\times 100\%$$
(2)$$\mathrm{Product}\;\mathrm{yield}\;\left(\%\right)=\frac{\mathrm{moles}\;\mathrm{of}\;\mathrm{formed}\;\mathrm{product}}{\mathrm{moles}\;\mathrm{of}\;\mathrm{initial}\;\mathrm{fructose}}\times 100\%\;$$
(3)$$\mathrm{Product}\;\mathrm{selectivity}\;\left(\%\right)=\frac{\mathrm{moles}\;\mathrm{of}\;\mathrm{formed}\;\mathrm{product}}{\mathrm{moles}\;\mathrm{of}\;\mathrm{fructose}\;\mathrm{converted}\;}\times 100\%$$

All insoluble and non-detected soluble products were considered as humins, i.e., they correspond to the detected C balance loss after carbon fraction distribution analysis and calculations. 

### 2.3. Recycling

For the recycling experiment, the reacted catalyst was recovered by filtration after each reaction cycle, washed with ethanol, and dried at 110 °C overnight before the next run. The reactive/catalyst ratio was maintained as constant in every run, according to the recovered catalyst mass.

### 2.4. Characterization Techniques

The textural properties of the samples were evaluated by nitrogen physisorption measurements in the Micromeritics TRISTAR II equipment. The material was degassed under vacuum at 150 °C for 12 h before analysis.

X-ray diffraction measurements were performed using a Panalitycal X’Pert Pro diffractometer, with a Cu anode (Cu-Kα 40 mA, 45 kV), using a step size of 0.05° and 300 s of step acquisition time within the 10–90° 2θ range. The Scherrer equation (Equations (4) and (5)) was applied to estimate the size of the carbon crystallites over the diffractions corresponding to the (002) and (100) family of planes:(4)$$L_{c}=\frac{K_{c}\lambda }{\beta_{002}cos\theta_{002}}$$
(5)$$L_{a}=\frac{K_{a}\lambda }{\beta_{100}cos\theta_{100}}$$
where *L_c_* is the crystallite height, *L_a_* is the crystallite diameter, *K* is the shape factor and depends on the crystal structure (0.9 for *K_c_* and 1.84 for *K_a_*), *λ* is the wavelength of the used X-rays (1.1584 Å for Cu anode), *β* is the full width at half maximum (FWHM), and *θ* is the position of the peak maximum.

Raman spectra of the different catalysts were recorded using a dispersive Horiba Jobin Yvon LabRam HR800 Confocal Raman Microscope equipped with a green laser (*λ* = 532.14 nm) and working at 5 mV power using a 600 grooves/mm grating and 50× objective with a confocal pinhole of 1000 μm.

The samples’ acidity was estimated over 100 mg of solid dispersed in 100 mL of distilled water using a pH-electrode (Metrohm, Oviedo, Spain). The pH values were collected after reaching a constant value.

SEM/EDX analysis of catalyst’ morphology and elemental composition was performed using a Hitachi S4800 SEM-FEG high resolution scanning electron microscope (Hitachi, Tokyo, Japan) provided with SE and BSE detectors and a Bruker X Flash Detector 4010 EDX analyzer (Bruker, Ettlingen, Germany) with a resolution of 133 eV for the Mn Kα line. 

The determination of nitrogen content was carried out using the elementary analyzer TRUSPEC CHNS Micro (Leco, St. Joseph, MI, USA), over 2 mg of sample at 1300 °C. 

Temperature-programmed desorption of NH_3_ was used to estimate the acidity sites’ strength and quantity. For analysis, 100 mg of sample was placed in a quartz reactor and submitted to a 50 mL/min He flow at 200 °C. After pretreatment, the sample was contacted with NH_3_ at 100 °C. The NH_3_ excess was evacuated in He at room temperature and subsequently heated to 500 °C with a 10 °C/min heating rate. The adsorption–desorption processes were followed by mass spectrometry using a Pfeiffer Vacuum (Aßlar, Germany) Prisma Plus Mass analyzer. Values of *m*/*z* = 16, 17, and 18 were registered and the ammonia evolution was considered as *m*/*z* = 17 minus the water contribution (corresponding to 26% of the *m*/*z* = 18 signal).

For the thermogravimetric analysis (TGA-DTG), the samples were dried at 110 °C overnight and analyzed from room temperature to 1000 °C (10 °C/min heating rate) under nitrogen flow (100 mL/min).

## 3. Results and Discussions

Table 1 summarizes the textural parameters of all samples. The observed hysteresis in N_2_ adsorption–desorption (Appendix A) indicate an IV H3 type isotherm, which according to the IUPAC suggests the presence of an important fraction of mesopores for all samples. In addition, the trend of the curve to infinity suggests also the presence of macropores. 

The commercial CNF sample shows a BET specific surface area of 160 m^2^/g and an average pore size of 15.5 nm. CNF treatment with sulfuric and nitric acid increases the specific area while all other functionalizations decrease it. In general, treatment with acids generates micropore and mesopore enlargement that accommodates the functional groups in the pores [38]; while the introduction of voluminous groups, such as organosilanes and ILs, causes the reverse effect and blocks access to the pores. The important decrease in BET surface area after the organosilane treatment is accompanied with an increase in pores size indicating a possible internal surface occupation with bulky molecules, responsible for micropore and low-diameter mesopore blocking. The latter is also justified by the complete absence of micropores for these samples and reduced single-layer adsorption quantities in comparison to those estimated for a mesoporous surface (Table 1). The oxidation of CNF-MPTMS to CNF-SO_3_HPTMS causes an increase in the BET area. This effect can be ascribed tentatively to the increase in the oxygen-containing groups, considering the SO_3_H groups formed by oxidation and all the newly formed hydroxylic and carboxylic groups and their active participation in nitrogen adsorption. 

CNF functionalization also affects the initial nanofiber structure (Figure 1). The XRD patterns present a sharp asymmetric main diffraction at around 2θ = 26.5° ascribed to the (002) planes of the multilayered nanographitic carbon crystalline domain with 3D graphite structure organization. This diffraction is accompanied by a broad signal in the 43–45° 2θ region related to the (100) and (101) family planes and two more diffractions observed at 54° and 78° 2θ corresponding to the rhombohedral graphitic structure.

As a general trend, the 002 planes’ diffraction increases after functionalization, suggesting higher graphitization, and the treated CNF samples gain a short-range order and become slightly more crystalline. Nevertheless, this order is not expanded to the long range and periodically repeated structures are not formed. This effect is more noticeable for acid- and ILs-treated samples. On the other hand, diffractions due to incorporated functional groups remain undetected, more probably due to the low initial loading of organic precursors onto the CNF surface. The average *in-plane* crystallite size (La) decreases for the acid-treated catalysts but increases after organosilane and ILs deposition (Table 2). Such an increase is due to the resulting larger carbon chains after covalent functionalization with organic precursors. On the contrary, the strong mineral acid treatments provoke carbon chain interruption, which results in a lower La parameter. The Lc parameters are very similar between the samples (Table 2).

The defect site population (analyzed by Raman spectroscopy and displayed in Figure 2) shows some changes after functionalization. The two intense bands at 1330 and 1590 cm^−1^, corresponding to *D* and *G* carbon vibration modes, dominate the spectra where the less intense 2D (*G*′) band at 2670 cm^−1^ is also visible. The untreated CNF sample shows a higher intensity *G* than *D* band and its calculated I(*D*)/I(*G*) ratio is much lower in comparison to the other samples, indicating a higher number of repeated *sp*^2^ hybridized graphene sheets (Table 2). During functionalization, the *D* band—and as a consequence, the calculated I(*D*)/I(*G*) ratio—increases, indicating a long-range order loss for the functionalized samples. It is worth mentioning the decrease in that ratio for CNF-APTMS and CNF-ILs, in comparison to their parent CNF-O sample, indicating a partial carbon sheets exfoliation (most probably the intersheet-stacking intercalation of the bulky molecules) and a possible *sp*^2^-structure re-ordering.

The morphological changes after functionalization are shown in Figure 3. One of the most apparent changes among the samples is CNF’s loss of packing after the treatments. The original fibers become shorter after the initial acid treatment (CNF-O or CNF-S) and maintain this distribution during the successive functionalization. The shortening of the fibers also reflects the increase in the specific surface area after the initial acid treatment. 

SEM-EDS and CNH elemental analysis help us to acquire the chemical composition of the catalysts (Table 3). The initial CNF sample contains sulfur (S), silicium (Si), and nickel (Ni) and presents a low degree of oxidation (O content). Silicium and sulfur are natural components of the CNF, but Ni is a possible residue from the catalyst used for nanofiber synthesis.

The composition changes correlate fairly well with the treatments. The CNF-O sample shows a higher oxygen content while CNF-S presents higher oxygen and sulfur contents. The incorporation of nitrogen, sulfur, and/or Si is indicative of the successful grafting of ILs or organo/mercaptosilanes. The organosilanes increase the total Si content, while the N and Cl contents rise with the introduction of ILs, with the parallel formation of BmimCl reported during the functionalization [39]. The lowest nitrogen and sulfur introduction observed for CNF-O and CNF-S in comparison to all other treatments indicates less carbon surface-functionalization agent interaction for these two samples. The latter is expected, as the obtained interaction does not proceed via covalent bonding. On the other hand, the covalent functionalization results in higher N and S loadings and therefore a higher population of different Brønsted/Lewis acid centers on the CNF surface. After thiol-to-sulfonic groups’ oxidation treatment (CNF-MPTMS to CNF-SO_3_HPTMS), the composition remains practically unaltered, with a slight increase in oxygen content due to the oxidation of thiol groups. Pertaining to Si content, the mercaptosilane functionalization seems more efficient than the aminosilane modification. 

The presence of all these superficial groups with different functionalities changes the comportment of the material in the aqueous phase, as confirmed by pH measurements. The introduction of amine groups, like in CNF-APTMS and CNF-ILs samples, shifts the pH to higher values, indicating a fast protonation of the superficial groups while the acid treatments decrease the pH to 4.84 for the CNF-SO_3_HPTMS sample (Table 2), suggesting the deprotonation of these Brønsted sites in water.

The total acidity of the samples, calculated by the integration of the NH_3_ desorption curves, is directly proportional to the amount of total acid sites, while the temperature of NH_3_ desorption indicates their strength (Table 2 and Figure 4).

The CNF sample presents a very low number of acid sites with a practically unaffected capacity to adsorb NH_3_ after nitric acid (CNF-O) treatment. On the contrary, all other functionalizations provoke a significant growth of the acid sites. The CNF-MPTMS and CNF-SO_3_HPTMS samples present two type of sites; (i) weak sites with a broad ammonia desorption in the 150–275 °C temperature range and (ii) stronger acid sites, with a desorption in the 275–500 °C region. Apparently, the functionalization of CNF with organic acids increases the fraction of stronger acid sites, while the treatment with sulfuric acid affects only the weak sites. On the other hand, the CNF-ILs and CNF-APTMS samples show the presence of a third type of site, although in a much lower amount for the former sample. A detailed analysis of the CNF-MPTMS and CNF-ILs broad contribution centered at 280 °C (*T*_max_) suggests the presence of medium-to-strong acidity centers. The highest proportion of sites for CNF-SO_3_HPTMS indicates that the post-oxidation process of the mercapto groups produces sulfonic entities of a much stronger acidity. 

Sulfuric acid and sulfur-containing organic functionalizations differ clearly in Brønsted acid site strength. The presence of Si and the propyl chain (for the organic functionalization) increases the strength of the sites, resulting in a higher contribution of moderate and strong acid sites. As a consequence, the hybrid CNF catalyst shows variable acidity sites associated either with a Lewis center (Si) as an electron pair acceptor or with pure Brønsted acid centers generated from the protons on the available surface groups.

The case of CNF-APTMS is somewhat surprising due to the important ammonia desorption observed at high temperatures. This contribution is understandable considering the amine–ammonia interaction of the amine group acting as a Brønsted center. This behavior can also be imagined as a consequence of the neighboring Si sites and HNHNH_3_ adduct formation, suggested by the protonation of the sample in water resulting in basic pH. We can also imagine the formation of a highly nucleophilic hydrazine radical (-NH_2_ + NH_3_ → N_2_H_4_ + H) in which the strong chemical bond formed is reflected in the higher desorption temperature [40,41,42].

The NH_3_ desorption areas normalized by the BET surface of the samples show a similar trend and can be related to the easy mesoporous surface access of all CNF-based catalysts.

The thermal stability of some catalysts in air is analyzed by TGA-DTG as summarized in Figure 5.

All samples show three temperature regions of weight loss, with the CNF-ILs sample being very different from the others. The first weight loss (below 200 °C) is usually ascribed to physically adsorbed water release and BmimCl melting (present in CNF-ILs). The (DTG) profile for CNF-ILs of this region (100–200 °C) shows two transitions, both assigned to adsorbed water and the second being delayed due to the difficulty of water discharge caused by the viscous character generated from the ionic liquid layer at 150 °C [43]. With the temperature increase, all functionalized samples show a slow weight loss in the region of 200–400 °C, more noticeable for CNF-ILs, due to the decomposition of functional groups bonded to the carbon skeleton. The sulfonated (oxygenated) samples show only one loss assigned to Brønsted groups’ decomposition and shifted to higher temperatures for the samples containing organosilanes. This step starts with the initiation of simultaneous graphitization of the samples according to the trend found in the literature [44]. From the final weight, it is possible to estimate the total weight loss and the percentage of mineral content and ashes. The nitric acid-treated sample shows almost complete weight loss, while the total mass present for SO_3_H-PTMS-CNF corresponds to the mineral content of the SiO_2_ component of the organosilane functionality. All catalysts present a good thermal stability to be used in dehydration reactions at temperatures below 200 °C. 

### 3.1. Biphasic Fructose Dehydration to HMF

The fructose dehydration reaction scheme over tandem (Brønsted–Lewis) acid catalysts is shown in Figure 1. The release of three water molecules to form HMF is mainly catalyzed by Brønsted acid sites, which are also responsible, at an excess Brønsted acid charge, for HMF rehydration to levulinic (LA) and formic acid (FA). At the same time, the presence of Lewis sites provokes fructose isomerization and undesirable condensation/oligomerization of hexoses to insoluble and soluble humins (also catalyzed by Brønsted sites). The lack of control on the strength and concentration of all incorporated Lewis/Brønsted sites could result in a catalyst being able to convert fructose rapidly to undesired products.

The catalysts’ screening is presented in Figure 6. The analyzed products are glucose, HMF, and levulinic and formic acid (lev + for), while the non-identified soluble or insoluble products are listed as humins. The biphasic MIBK/H_2_O system consists of two immiscible phases, allowing fast distribution of the formed HMF between the organic and aqueous phase, while all other products (acids or sugars) remain in the aqueous phase. The partition coefficient (organic/water) for HMF is 1.19 for an MIBK/H_2_O ratio of 5/1.

The use of a biphasic system seems to prevent HMF rehydration to levulinic and formic acids with their fraction being below 6% in all cases. One can imagine that MIBK participates indirectly by suppressing the side reaction of HMF rehydration. The appearance of glucose, as a product, indicates fructose isomerization reactions. The CNF-ILs catalyst shows the highest activity in this reaction followed by CNF-MPTS and the unmodified CNF structure. This activity can be tentatively attributed to the presence of heteroatoms within the *sp*^2^ aromatic sheets like in imidazolium, chromenes, and pyrones [45,46].

The higher hydrophobicity and lower oxygen content and acidity of the CNF sample can explain its low conversion (23%). Upon functionalization, no matter the treatment, fructose conversion increases. In general, the N-containing functional groups allow for higher conversion than for S-containing samples. Within the N-containing group, the stronger acidity of the CNF-ILs and CNF-APTMS functionalized catalysts is reflected in a higher fructose conversion (73% and 90%, respectively), while the weak sites of the CNF-O sample result in a lower conversion (46%). 

The HMF yield seems to be influenced by catalysts’ structural and textural properties. The CNF-O sample shows a very low HMF production (6%) in comparison to unmodified CNF (17%). The slight increase in acidity and the pore volume increase for CNF-O potentiates the cross-polymerization of hexose’s tautomers, leading to significant humin formation in detriment to HMF. 

Apparently, the stronger acidity of CNF-APTMS and CNF-ILs samples causes important byproducts’ generation either by HMF or glucose polymerization reactions. While the base groups of CNF-APTMS (NH_2_ groups) coupled with Lewis sites increase the direct HMF polymerization, CNF-ILs is more selective to Lobry De Bruyn Van Ekeinstein fructose-to-glucose isomerization and subsequent participation in glucose self- or cross-polymerization to humins [10,47]. The grafted imidazolium plays a dual acid/base role, participating in both isomerization and dehydration, whereas the compensating chlorine ions act as nucleophiles that bind fructose through H-bonds, thus stabilizing the intermediates and transition states to finally minimize HMF and humin production [48].

On the other hand, sulfur-containing samples significantly improve the HMF production with a 62% HMF yield in the best case (CNF-SO_3_HPTMS). The HMF yield appears to be related to the presence of Brønsted sites like -SH and -SO_3_H groups (CNF-SO_3_HPTMS, CNF-SO_3_HPTMS and CNF-S) and fructose/functionalized CNF surface interaction. The presence of bulky grafted molecules (CNF-SO_3_HPTMS, CNF-SO_3_HPTMS) introduces a certain hydrophobicity, pushing the solids to the biphasic interface and weakening the interfacial tension. The latter facilitates the protonation of furanose and faster production/extraction of HMF. The combination of both the higher acidity of the moderate type and faster HMF transfer makes the CNF-SO_3_HTPMS sample the best catalyst for this reaction in terms of conversion and selectivity.

The correlation between the reaction parameters (fructose conversion, HMF yield and selectivity, and humin selectivity) are plotted as a function of catalyst acidity in Appendix A. Within the series, the fructose conversion seems to follow the order of acidity. The higher the overall acidity, the greater the conversion. The HMF yield and selectivity show the same behavior, with the CNF-APTMS sample out of range due to the important parallel undesired reactions. Excluding this sample, we can conclude that in general, the increase in the acid fraction improves HMF selectivity and production. The distribution of humins and its relationship with samples’ acidity appears random (Appendix A). While CNF-APTMS presents the highest humin production (and high acidity), the CNF-SO_3_HPTMS sample shows the inverse trend. It seems that the high acidity of the CNF-SO_3_HPTMS does not result in high humin production, probably caused by the presence of Si as a modulator of the strength of the Brønsted sites.

The catalytic performance in fructose dehydration in a biphasic system is not defined only by the nature of the catalyst but also by other parameters like the solvent ratio, temperature and time of reaction, and initial fructose concentration. All catalytic parameters are optimized toward the maximum HMF yield using the best catalyst, CNF-SO_3_HPTMS. The results, summarized in Figure 7, suggest that the MIBK/H_2_O ratio does not affect the fructose conversion but the HMF production. The highest yield (64%) is obtained for an MIBK/H_2_O ratio of 5/1. At higher ratios, a slight HMF yield decrease is observed due to the higher formation of humins. Nevertheless, this decrease is as significant as the decrease observed at low MIBK/H_2_O ratios (3/1). In general, MIBK also participates in the reaction acting as a furanose tautomer stabilizer, which is more favorable for HMF production [49]. However, a continuous increase in solvent quantity leads to a weaker polarity of the solvation system and to a general decrease in the proton concentration available in the reaction media and needed for fructose dehydration to HMF. Therefore, an optimal ratio of 5/1 is observed for a maximal HMF yield.

The effect of the reaction time on catalysts’ dehydration behavior was measured at a constant temperature of 165 °C using a 5/1 solvent ratio (Figure 8). In the first 30 min, the fructose conversion and HMF yield reached 21% and 13%, respectively. Fructose conversion increases with time, while the HMF yield shows a maximum at 240 min and then drops. At longer reaction times, it is difficult to improve the HMF yield as the condensation reaction occurs and humins start to appear. Therefore, the reaction time of 240 min is selected as the optimal time of reaction.

Figure 9 shows the results obtained as a function of temperature at 240 min of reaction and a solvent ratio of 5/1. Fructose conversion increases continuously with the temperature up to 89% at 185 °C. The HMF yield shows a maximum of 62% at 165 °C, and afterwards a higher tendency to self-condensation and polymerization appears.

Thus, the effect of initial fructose loading is evaluated at 165 °C as the optimal temperature. Changing the initial fructose loading from 90 mg to 360 mg negatively affects the conversion, indicating an insufficient number of available active sites (Figure 10) at higher substrate loading. The maximum HMF yield and selectivity is obtained at 180 mg of substrate (72% conversion and 62% HMF yield). 

A higher substrate/catalyst ratio decreases the HMF yield. In fact, it decreases the accessibility to the active sites, increases the residence time on the sites, and increases the probability of secondary reactions, thus diminishing the HMF yield. 

### 3.2. Catalyst Reuse

The recyclability is an important advantage of the solid catalyst over its homogeneous analogs. CNF-SO_3_HPTMS catalyst is used in five reaction cycles and the corresponding results are included in Figure 11. The catalytic performance remain stable in five consecutive cycles. This behavior is expected in view of the adequate covalent functionalization of the CNF surface and optimal reaction conditions. The leaching is virtually impossible and the used catalyst appears as an excellent candidate to be used as heterogeneous catalysts for efficient HMF production from fructose.

### 3.3. Catalyst Kinetics and Tentative Reaction Mechanism on Functionalized CNF

In accordance with the experimental results and the literature [16,29,50,51,52,53], we fit the kinetics of HMF production to a first-order reaction of fructose transformation to HMF and humins proceeding via common intermediates.

The results of kinetic model for fructose conversion, calculated by Equation (6) are plotted in Appendix A.
(6)$$-\ln\;\left(1-x\right)=k\times t$$
where x: fructose conversion, k: rate constant (min^−1^), and t: reaction time (min).

The first linear approximation with fixed first order shows a very high concordance with the experimental data with a calculated constant rate value of 0.00213 min^−1^ at 165 °C and a great correlation coefficient factor (R^2^ = 0.9924), suggesting plausible initial assumptions. 

Thus, the validated equation was used further for the determination of the rate constant at different temperatures (Appendix A) and apparent activation energy Ea calculation for fructose dehydration to HMF in the presence of the CNF-SO_3_HPTMS catalyst. A five-fold increase in the constant is found within the studied range of temperatures indicating a reaction acceleration at higher temperatures. The activation energy Ea is calculated from the slope of the plot (Appendix A and Appendix A).
$$-\ln\;\left(k\right)=\ln A-\frac{\mathrm{Ea}}{R}\;\frac{1}{T}$$
with k: rate constant (min^−1^), *A*: pre-exponential factor (min^−1^) R = 8.314 × 10^−3^ (KJ mol K^−1^), *T*: reaction temperature (°K), and Ea apparent activation energy (KJ mol^−1^).

The calculated Ea shows that the amount of adsorbed fructose is not exactly correlated with the temperature change, and therefore the activation energy should have been higher than the heat of adsorption. The Ea for the CNF-SO_3_HPTMS catalyst (21.8 KJ/mol) is much lower than other studied catalysts, like niobium and zirconium phosphates in water (65 and 186 KJ/mol respectively) [54,55], bi-functionalized mesoporous silica (67.5 KJ/mol) [56], and halloysite-supported silicotungstic acid (85 KJ/mol) in DMSO [57]. 

These catalytic results and kinetics over the CNF-SO_3_HPTMS catalyst indicate very good behavior in HMF production from highly concentrated fructose solutions, converting the process to be more environmentally friendly with a low energy consumption suitable for industrial applications.

As for the mechanistic behavior, clearly the dehydration of fructose to HMF is related to the Lewis /Brønsted acidity strength and ratio [58,59,60,61]. Both type of sites participate in fructose conversion. While Lewis sites catalyze the tautomerization of the enol intermediate to aldehyde (Step 2 in Figure 2), the Brønsted sites are majorly involved in the decrease in the HMF production energy barrier. On the other hand, both centers catalyze the protonation of OH groups, with Brønsted sites providing the protons and Lewis centers participating in the lone-pair electron OH groups’ attack using their empty metal orbital (Steps 1, 3, and 4). 

After adsorption and protonation, the fructose cannot dehydrate in the presence of Brønsted sites only (like in the case of CNF-O and CNF-S) and the hexose tautomer remains in the pores of the catalysts to further oligomerize to humins. The presence of a nucleophile such as a Lewis base (CNF-ILs) is needed to prevent the humin formation. On the other side, the presence of a Lewis acid coupled with a Lewis base, as in the case of CNF-ILs and CNF-APTMS, leads to low HMF production but high glucose production. 

To improve the HMF production and to inhibit humin formation, the presence of Lewis acid sites and weak to medium Brønsted acid sites (as in the case of sulfur organic acid containing CNF, CNF-SO_3_HPTMS, and CNF-MPTMS) is desirable. It must be underlined that an optimum amount and ratio of acid sites are required for successful fructose dehydration over CNF-based catalysts. Contrasting with the NH_3_-TPD analysis, we can conclude that an important number of acid sites with moderate strength is required to improve HMF production and inhibit secondary reactions.

## 4. Conclusions

Novel carbonaceous hybrid catalysts, based on functionalized nanostructured CNF with amino- and sulfur-containing organosilane and ionic liquids entities, have been successfully prepared by covalent grafting. The latter allows the introduction of tandem Bronsted/Lewis acid sites of a different nature, strength, and number. The type of acid centers affects the catalytic properties of the functionalized materials to a great extent. The best catalyst, CNF-SO_3_HPTMS, presents a fine balance between weak and moderate Brønsted sites coupled to Lewis sites, resulting in an excellent catalytic choice in terms of fructose conversion and HMF yield. Further, the fructose dehydration can be tuned by changing some reaction variables, with a higher MIBK/H_2_O ratio being more beneficial than an increase in time or temperature. The outperforming CNF-SO_3_HPTMS catalyst is also stable under various reaction cycles, allowing us to conclude that a significant advance in the preparation of novel bi-functional hybrid organic–inorganic material by the covalent grafting method is achieved and opens the possibility to produce a new generation of multifunctional solids for biomass valorization.

## Data Availability

The raw/processed data required to reproduce these findings cannot be shared at this time as the data also form part of an ongoing study.

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
