# Peer review of "Finely Tunable Carbon Nanofiber Catalysts for the Efficient Production of HMF in Biphasic MIBK/H2O Systems"

_nanomaterials, 2024, doi:10.3390/nano14151293_

Round 1

Reviewer 1 Report

Comments and Suggestions for Authors

In the present manuscript titled as “Finely tunable CNFs catalysts for the efficient production of HMF in biphasic MIBK/H2O systems” Bounoukta et al. prepared a series of CNF catalysts with different functionalization and shows their impact on fructose dehydration to produce HMF The results shown here are interesting and can be considered for the publication after authors carefully revised the following comments. Significant improvement is also required for the discussion section.

Please find below the comments and suggestions that needs to be addressed in detail.  

1.       Why CNF-APTMS has large fraction of strong acid sites compared to CNF-SO3HPTMS. It is apparent that amine functionalization incorporates more base sites compared to acid sites

2.       TPD-NH3 area reported in Table 2 corresponding to Figure 3? If this is the case then area corresponding APTMS samples looks much larger than any other materials but this is not the case according to the value reported as well as the description in text at page 9.

3.       The discussion about the catalytic activity and selectivity needs to improve significantly. It is very confusing at present condition.

4.       Why presence of basic functionality induces higher fructose conversion. If the dominant mechanism is dehydration as shown in scheme 1 then it should have a correlation with number of acid sites.

5.       How authors define activity of catalysts? Is it fructose conversion? Authors mentioned that CNF-SO3HTPMS is most active and selective. However, fructose conversion is higher in case of APTMS catalyst.

6.       In the present case typical substrate to catalyst ratio is ~4 which apparently very low compared to traditional catalysis. Did authors make sure that these results are free of diffusion limitation. Indeed, low activation energy obtained here compared to other report also suggest the possibility of diffusion limitation.

7.       Based on Figure 6, the highest selectivity to HMF was obtained with SO3H functionalized catalyst, the best one studied herein, however the value is very similar to baseline CNF. Why there is no impact on selectivity although acid/base sites are very different in these two catalysts?

8.       The activity trend discussed herein is significantly influenced by the nature of acidity or basicity. Authors should show some figure corelating activity/selectivity to functionalization. Based on the discussion there should be some volcano type relation.

9.       The trend shows in figure 10 is not very clear. Why there is a steady drop in fructose conversion with increase in substrate loading? Is there an inhibiting effect from substrate? This is a batch reaction so I do not think residence time is a factor here.

Comments on the Quality of English Language

Overall English language is fine but discussion section especially where authors discussed the performance of the catalysts needs to be improved significantly as it is difficult to follow.

Reviewer 2 Report

Comments and Suggestions for Authors

In this contribution, authors present and propose the use of a series of CNFs-based catalysts for the for the fructose dehydration to 5-hydroxymethylfurfural.

The preparation and physico-chemical properties of the synthesized catalysts are well-explained, with the CNF-SO3HPTMS catalyst exhibiting the best performance in catalytic tests. The authors thoroughly elucidate the properties of the catalysts and their activity. The experimental section is rich in data, and the results are interesting.

In general, the paper is solid, and the conclusions are consistent with the experimental data. Therefore, the manuscript may be of interest for publication in the nanomaterials journal, but the following points need to be addressed before publication:

-       On page 3, equation (3) is indicated as “Product Selectivity,” while throughout the text the authors refer to “Yield.” The authors should correct this.

-       Figure 6 should be improved. It would be better to combine the images into a single figure.

-       Some English polishing to correct typos and mistakes is needed.

-       Some references need to be formatted correctly.

Comments on the Quality of English Language

Some English polishing to correct typos and mistakes is needed.

Reviewer 3 Report

Comments and Suggestions for Authors

This paper presented some interesting results in HMF synthesis. However, I have two main points the Authors shoud address before the paper can be accepted for publication.

1) acidic site are fundamental but they are not considered very much in the explanation. In my opinion they have to be classified in strenght and, most importantly, normalized with surface area. From TPD-NH3 for example it is clear CNF-O and CNF-SO3HPTMS present same area and same pH but behave very differently.

2) most studies use glucose as model starting material. Here fructose is used. Why? What about isomerisation reaction?

3) Ni impurities have been found. What is the possible impact of basic NiO on the reaction?

Comments on the Quality of English Language

English is OK

Round 2

Reviewer 1 Report

Comments and Suggestions for Authors Authors provided sufficient explanation to all of the questions and the manuscript can be accepted for the publication in current form. Comments on the Quality of English Language

Overall English language is fine.